# Trophic ecology in an anchialine cave: A stable isotope study

**Brenda Durán**[1,2][☼], **Fernando Álvarez**[2][☼]*

1 Posgrado en Ciencias Biológicas, Universidad Nacional Autónoma de México, Ciudad Universitaria, Mexico, Ciudad de México, Mexico, 2 Colección Nacional de Crustáceos, Instituto de Biología, Universidad Nacional Autónoma de México, Ciudad Universitaria, Mexico, Ciudad de México, Mexico

☼ These authors contributed equally to this work.
* falvarez@unam.mx

**Data Availability Statement:** All relevant data are within the manuscript and its Supporting information files.

**Funding:** BD CONAHCYT graduate scholarship FA grant PAPIIT IN206523 (DGAPA-UNAM) FA grant CONAHCYT Ciencia Básica A1-S-32846. The

## Abstract

The analysis of carbon and nitrogen stable isotopes ($\delta^{13}$C and $\delta^{15}$N) has been widely used in ecology since it allows to identify the circulation of energy in a trophic network. The anchialine ecosystem is one of the less explored aquatic ecosystems in the world and stable isotope analysis represents a useful tool to identify the routes through which energy flows and to define the trophic niches of species. Sampling and data recording was conducted in one anchialine cave, Cenote Vaca Ha, near the town of Tulum, Quintana Roo, Mexico, where seven stygobitic species endemic to the anchialine caves of the Yucatan Peninsula, plus sediment, water and vegetation samples were analyzed to determine what the main nutrient sources are. We compared our results with two previous studies, one conducted in the same cave and another one from a cave in the same area, both based on the same seven species which are widely distributed in the area. Our study revealed: a) that despite a certain amount of variation in the $\delta^{13}$C and $\delta^{15}$N values of the species through time, both seasonally and interannually, the anchialine isotopic niche is much conserved; b) through contribution models we propose what are the most probable food sources for the studied species and the results confirm previous trophic classifications; and c) that the shrimp *Typhlatya pearsei* presents very negative $\delta^{13}$C values, suggesting their consumption of bacterial sources consistent with a chemosynthetic origin of organic matter. The implications of the new findings show a very stable ecosystem with the shrimp *Typhlatya pearsei*, as the key species to link chemosynthetic microbial production of organic matter to the anchialine trophic web.

## Introduction

Stable isotope analysis (SIA) has significantly contributed to identify the major routes through which energy flows in different ecosystems and has been an important tool to define trophic structure and to identify the number of trophic levels in a given ecosystem [1, 2]. In hard to access ecosystems where sampling is technically difficult, such as the flooded caves of the anchialine ecosystem [3], SIA offers a way to explore the structure of the trophic web through

funders had no role in study design, data collection and analysis, decision to publish, or preparation of the manuscript.

**Competing interests:** The authors have declared that no competing interests exist.

the analysis of samples of sediment, water, organisms, and the surrounding vegetation. The data obtained through this approach can be complemented with occasional and/or opportunistic observations, or through laboratory experiments [4]. In order to follow a logical sequence, we need to know what species occur in the anchialine habitat and what their likely trophic roles are, how nutrients are entering the caves and what the possible sources are, and finally under what degree of variation–spatial and temporal–is this structure operating?

Anchialine ecosystems have a rich and endemic fauna around the world that typically thrive in low nutrient conditions [5]. With a reduced rate of transport of nutrients into the flooded caves [6], the recurrent question has been, how can a diverse fauna with some locally abundant species develop and survive under these conditions? Specially in the Yucatan Peninsula (YP), where flooded caves can form networks spanning hundreds of kilometers [7], the conditions and mechanisms that shape the trophic web operate at various scales. There could be a short distance nutrient input that is present in the cave area adjacent to openings, such as the cenote pool, or cracks that connect the exterior to the aquifer; however, in remote areas inside the caves, hundreds or even thousands of meters from an opening to the exterior, other mechanisms may be operating differently from the direct entry of organic material from the exterior. Thus, identifying the main nutrient sources and the major paths through which energy flow becomes of primary importance.

Anchialine caves around the world are characterized as containing oligotrophic to ultraoligotrophic water masses, a condition that has promoted the appearance of both morphological and physiological adaptations in the fauna to cope with low nutrient concentrations [5, 8]. Nutrient input to the flooded caves of the YP has been associated to rainfall regimes and to the distance of the particular cave section to the shoreline [9]. However, even when seasonal fluctuations occur these cave systems remain nutrient poor [10].

Previous studies of the trophic structure of anchialine communities using SIA [11–16] identified a common pattern where anchialine fauna have $\delta^{13}C$ and $\delta^{15}N$ values that reflect the influence of the vegetation surrounding the cave entrance, in the YP these dolines or exterior connections to the aquifer are known as "cenotes". Derived from these data short trophic webs have been described, with a first trophic level composed of filter feeders that consume particulate organic matter or grazers that feed on the biofilm and a second level composed of predatory species [11]. In the caves of the YP the first trophic level includes species such as the mysid shrimp *Antromysis cenotensis* which typically occurs near the caves' entrances or even in the twilight zone and feeds on finely particulate organic matter [17] or the shrimps of the genus *Typhlatya* which possess a feeding apparatus made up of specialized setae to graze on the bacterial mats [18]. The species that compose the second trophic level are opportunistic predators such as the remipedes of the genus *Xibalbanus*, the brotulid fish *Typhlias pearsei* or the palaemonid shrimp *Creaseria morleyi* which have been observed preying on *Typhlatya* shrimps [5, pers. obs.]. Another significant component of the trophic web in the anchialine caves of the YP is the isopod *Creaseriella anops*, one of the most abundant organisms in this ecosystem, which acts as a scavenger that rapidly consumes all dead animals and decaying debris [19].

Despite the entry of organic matter to the flooded caves from the outside is evident, at least in passageways adjacent to connections to the exterior, in several instances direct observations and SIA have identified a second possible source of nutrients. In this case the degradation of organic matter by nitrifying bacteria and, depending on the type of cave and available dissolved compounds, the use of methane by chemoautotrophic bacteria to produce organic matter can supply additional nutrients [11, 12]. An important bacterial and archaea diversity has been identified in cenotes and caves of the YP with a clear zonation according to the distance to the coastline and thus the influence of saltwater penetration that creates different

geochemical conditions [20]. The overall contribution of bacterial activity to the anchialine trophic web is becoming increasingly relevant as exploration advances and the enormous lengths of the caves in the Yucatan (in the hundreds of kilometers) pose doubts about the effectiveness of water circulation to spread nutrients throughout.

In a few instances, very negative $\delta^{13}C$ values (<-36‰) have been recorded in the YP caves obtained from crustaceans in general or shrimp that are consistent with their feeding on nitrifying bacteria [13, 15]. Building on these ideas other authors [21] showed that a chemosynthetic route can operate under particular conditions inside anchialine caves, providing a possible explanation for the impoverished $\delta^{13}C$ values that have been obtained [16]. Previously [11] it was hypothesized that one thermosbaenacean, *Tulumella unidens*, and the atyid shrimp *Typhlatya mitchelli*, could be feeding on sulfide-oxidizing and methanotrophic bacteria.

Recent studies using a variety of techniques have identified a chemosynthetic route through the degradation of methane by bacteria to produce organic matter in an anchialine cave near Tulum, Quintana Roo, Mexico [21, 22]. With these important contributions to the understanding of the anchialine trophic web, several critical questions arise. Firstly, considering that there are thousands of flooded caves in the YP, how widespread is this mechanism? That is, the very negative $\delta^{13}C$ values can be found in all caves in the YP? Secondly, what species are involved in this process and how can their distribution affect such biogeochemical process? Third, do these processes influence the distribution of species creating, where they operate, hotspots of diversity?

Another relevant question regarding the stability of the anchialine trophic web is if it is influenced by seasonal (rainy vs. dry seasons) or interannual changes due, for example, to the large-scale impacts of deforestation or the occurrence of hurricanes. Since SIA reflects what organisms have incorporated in the previous weeks or months, it should be possible to detect these changes to elucidate how the anchialine ecosystem reacts to them. In this study, based in a cenote and its associated flooded cave that is representative of the cave area that develops along the Caribbean coast of Mexico (Fig 1), we compare the $\delta^{13}C$ and $\delta^{15}N$ values of seven anchialine species (*Antromysis cenotensis*, *Stygiomysis cokei*, *Tuluweckelia cernua*, *Creaseriella anops*, *Typhlatya mitchelli*, *T. pearsei*, *Xibalbanus tulumensis*), sediment and organic matter in one cave in two seasons (rainy and dry). To explore variation among years, we compare also the new results with the values of a previous study [11], obtained from a cave very close to our study site and based on the same species, and from the cave identical with the present study, obtained 2 years before the recent study [23].

## Materials and methods

The study was conducted in Cenote Vaca Ha and its associated cave, near the town of Tulum in the State of Quintana Roo, Mexico. Cenote Vaca Ha (20º16'14" N, 87º28'49" W) is located 6.67 km from the coastline, the pool is completely exposed and has an area of ~25 m$^2$ (Fig 1). Cenote Vaca Ha is surrounded by vegetation, mostly grasses and perennial trees, some palm trees are found in the surrounding areas far from the cenote's water body, in the rainy season a large swamp area develops that connects to the cenote pool. The site is only used by divers. The cenote was named "Vaca Ha" (cow water) because the land where it was discovered was used as a livestock maintenance site, taking advantage of the cenote pool as a watering hole. Currently the land is private, and divers pay an entry fee. The interior of the cave contains wide passages, enormous columns and a rich decoration of speleothems, the passage reaches a depth of 24 m where it is possible to observe the halocline.

Despite the continuous exploration of flooded caves in the area, and the presence of two of the largest cave systems in the world (Ox Bel Ha and Sac Actun) at a very short distance

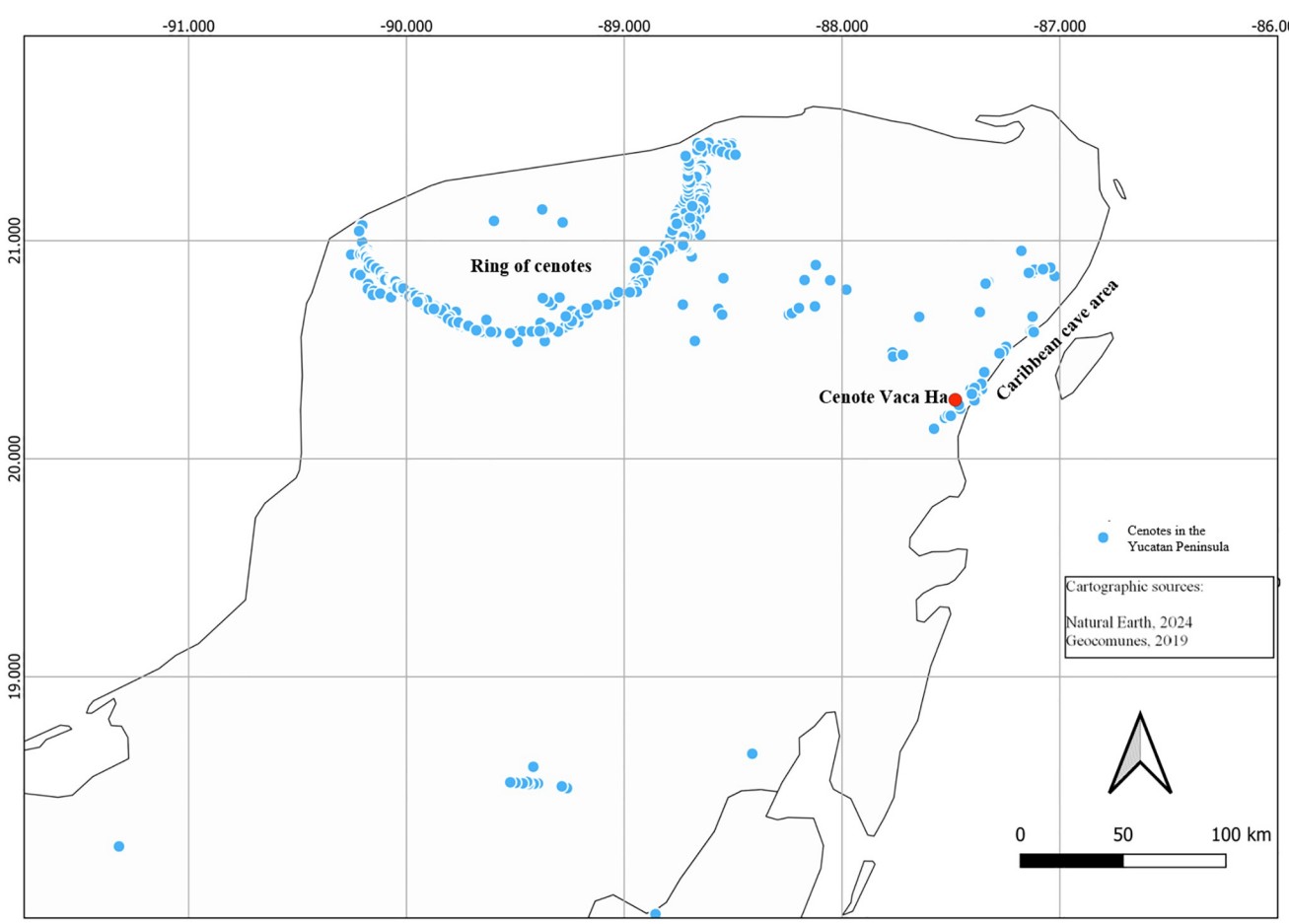

**Fig 1. Map of the Yucatan Peninsula, Mexico, with cenotes shown in blue circles, and Cenote Vaca Ha indicated by a red circle.** The map image was obtained at Natural Earth (https://www.naturalearthdata.com/) which is in the public domain. The layer with cenotes is taken from GeoComunes portal, and is in the public domain (http://132.248.14.102/layers/CapaBase:cenotes_completo2).

(< 10 km), Cenote Vaca Ha and its associated cave remains as an unconnected independent system. Notwithstanding the absence of known connections, there is obvious water circulation inside the cave.

## Field work

Sediment samples were collected in 50 ml Falcon tubes, water in collapsible 5-liter plastic bags, and stygobitic fauna individually in small vials. Collection of organisms was conducted inside the cave, starting 15 m from the cenote in complete darkness, under permit SAGARPA PPF/DGOPA-084/22. In addition, vegetation samples (leaves) from the rainforest floor in a radius of 50 m around the cenote were also collected randomly with tweezers. The common plants and trees in the area were: *Erythrina standleyana*, *Gliricidia sepium*, *Brosimum alicastrum*, *Ceiba aesculifolia* and *Casearia emarginata*. All samples were placed in sterile plastic vials. A mixture of 50 gr of leaves was dried to obtain one sample. Water column profiles were obtained with a YSI EXO 3 sonde programmed to take readings every 15 seconds of: depth (m), temperature (ºC), pH, conductivity (mSc), salinity (ppt) and dissolved oxygen concentration (mg/L).

Samplings were conducted by two cave divers in July 2022 for the rainy season and in March 2023 for the dry season. Every dive lasted approximately 1 h; in case some of the samples or data were missed, a second dive was done the next day to complete the sampling.

## Sample processing for carbon content analysis in water and sediments

To estimate the total dissolved carbon, organic and inorganic, 600 ml of water were filtered using pretreated (ignited at 500 ºC for 4 h) glass fiber filters (0.7 μm); from these three 100 ml water subsamples were collected in amber vials. Water samples were taken in the cenote pool and in the cave, in the cave two types of samples were taken, one in the freshwater layer at 8 m depth and one in salt water at a depth of 23 m. These samples were analyzed at the Laboratorio Universitario de Nanotecnología Ambiental (LUNA), of the Instituto de Ciencias Aplicadas y Tecnología (ICAT), UNAM using a Total Organic Carbon Analyzer (Shimadzu TOC-L CSH/CSN).

To obtain the amount of total carbon (organic and inorganic) present in the sediment of the cenote pool and in the cave´s interior, samples were taken with Falcon tubes. Five cave sediment samples were taken, each one 15 m apart starting the first sample in the cenote pool (Fig 2). The samples were dried at 60 ºC for 48 h and analyzed at LUNA of ICAT-UNAM using the same Total Organic Carbon Analyzer as above but with the adapter for solid samples SSM-5000a.

## Sample processing for SIA

Sediment samples were treated with hydrochloric acid (HCl) before they were analyzed to eliminate all carbonates following a conventional technique [24]. The sediment and vegetation samples were dried at 60 ºC for 48 h, and then grinded until a fine powder was obtained.

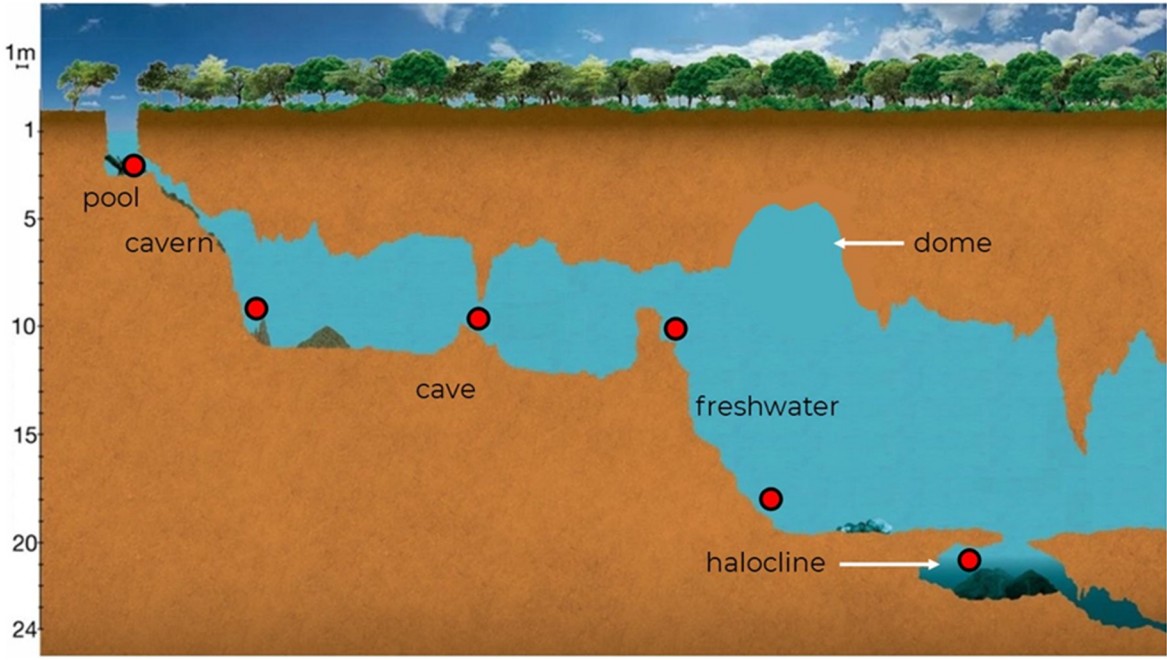

**Fig 2. Schematic representation of Cenote Vaca Ha showing the different sections of the system, the two main water masses and the sites where water samples were taken for carbon measurements.**

Seven species of crustaceans collected were maintained live in 500 ml plastic containers with filtered (0.45 µm) cave water and aeration for 24 h. During this period the organisms would empty their digestive tracts, and water was changed once to avoid coprophagy [25]. The organisms were then placed in 50 ml sterile vials and frozen at -4˚C; the samples were taken to the laboratory, defrosted at room temperature and dried at 60 ºC for 48 h, and then grinded until a fine powder was obtained. When organisms were small (e.g., mysid shrimps, amphipods) more than one organism was used to comply with the minimum weight required for one measurement of the isotopic analysis. With larger organisms (isopods, atyid shrimps, palaemonid shrimps, remipedes) it was possible to prepare one sample from each individual.

All samples, placed in tin capsules, were analyzed in a mass spectrometer (Isotopic Ratio Spectrometer, Isoprime 100TM, CF-EA) at the Stable Isotopes Laboratory of the Geotop Center at UQAM (Université du Québec à Montréal, Montréal, Québec, Canada). Carbon and nitrogen isotope ratios are expressed as $\delta^{13}$C to Vienna Pee Dee Belemnite standard and as $\delta^{15}$N to air.

### Data analysis

The $\delta^{13}$C and $\delta^{15}$N values, that is the means for every species, for the dry and rainy seasons were compared directly for each species. To graphically represent the isotopic values of the proposed sources as well as those obtained in the tissue of the consumer species (average values and their standard deviation), we constructed Bayesian mixing models (BMMs) with the MixSIAR R package using $\delta^{13}$C and $\delta^{15}$N as isotopic tracers to estimate the proportional contribution of potential food sources to their diets [26], this package is included in the R software [27]. We used the classic TDF 0.5‰ for $\delta^{13}$C and 3.4‰ for $\delta^{15}$N. The models obtained estimate the proportional contribution of the proposed sources considering the values of the isotopic signal of the consumers and the existing isotopic fractionation. The results are presented as credibility intervals (50, 75 and 95) that describe the range of possible contributions from a particular source (see Fig 5). In this study the use of BMMs is particularly advantageous since the number of species involved in the anchialine trophic web is relatively small and there are one or two trophic levels [28].

To explore how the isotopic niche of the studied species changes through time we used the package SIBER in R to create scatter diagrams of the carbon and nitrogen data and independent ellipses, representing the isotopic niche [29], for each of the analyzed datasets: one from 1997 [11], another one from 2020 [23], and the rainy and dry seasons presented herein for 2022–2023. We used the standard ellipse area (SEA) method to compare between the analyzed groups which shows the 50, 75 and 95% credibility intervals (see Fig 6).

## Results

### Physico-chemical characterization of the water column

The physico-chemical sonde profiles show a stratified water column with a well-defined halocline at around 21 m, overlapping values are due to the presence of a shallow dome inside the cave which is shallower than the bottom of the cenote's pool. At the halocline salinity changed from 2 to 35 in less than a meter (Fig 3A). Dissolved oxygen concentration was close to 4 mg/L at the pool's surface, it dropped to less than 0.5 mg/L inside the cave's dome, was constant around 3 mg/L in the freshwater layer and then dropped again to less than 1 mg/L in the saltwater (Fig 3B). Temperature was near 25.9 ºC at the pool's surface, slightly decreasing to 25.7 at the pool's bottom, it increased about have a degree inside the cave´s dome and then remained constant down to the halocline where it slightly increased (Fig 3C). Finally, pH was about neutral throughout the freshwater layer, except in the cave´s dome where it slightly

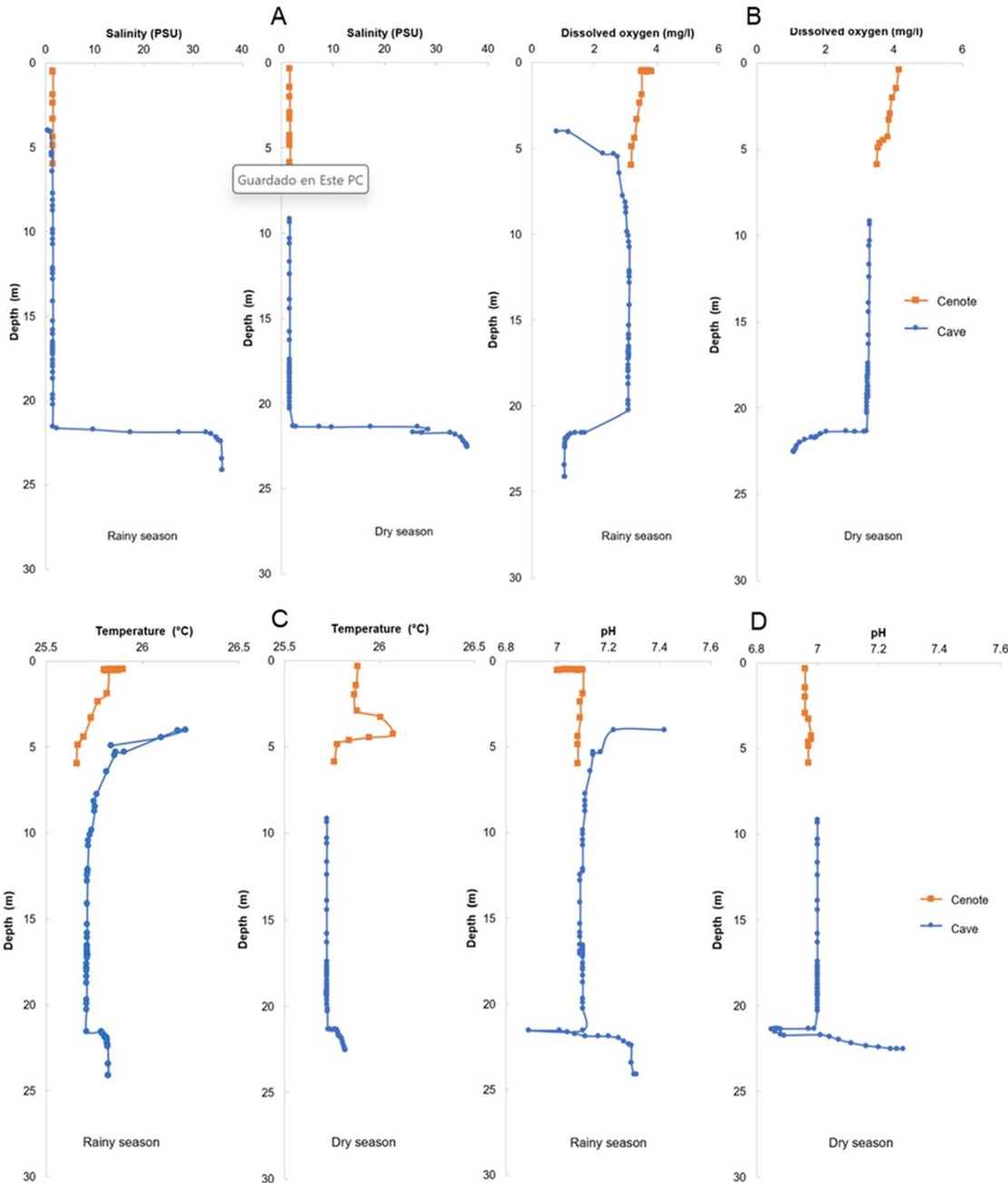

**Fig 3. Physico-chemical profiles of the water column in Cenote Vaca Ha for the rainy and dry seasons.** (A) salinity, (B) dissolved oxygen concentration, (C) temperature, (D) pH.

increased 3 tenths of a unit, it decreased within the halocline and recovered in the saltwater layer (Fig 3D). In general, no important variations are observed between the rainy and dry seasons in any of the four variables measured.

Total dissolved carbon was highest in the cave's freshwater layer, decreasing slightly towards the cenote pool and decreasing sharply in the saltwater layer (Table 1). Organic carbon represented 90.5% of all the dissolved carbon, and both organic and inorganic carbon had

**Table 1. Dissolved carbon in three sections of the aquifer in Cenote Vaca Ha, the samples had a volume of 100 ml.** TC = total dissolved carbon, OC = dissolved organic carbon, IC = dissolved inorganic carbon.

|  | Cenote pool | Cave (fresh water) | Cave (salt water) |
|---|---|---|---|
| TC | 81.82 mg/L | 96.01 mg/L | 41.53 mg/L |
| OC | 74.04 mg/L | 87.43 mg/L | 36.46 mg/L |
| IC | 7.78 mg/L | 8.58 mg/L | 5.07 mg/L |

the same behavior regarding concentration as the total organic carbon (Table 1). Total carbon in sediments decreased steadily along the transect that started in the jungle floor, through the cenote pool and inside the cave to a penetration of 75 m. The amount of total carbon decreased to 38.7% of what was available in the exterior and the organic and inorganic fractions varied inversely, with the former being more abundant in the exterior and cenote pool, and the latter comprising almost all the available carbon inside the cave (Table 2).

## SIA analysis

Stable isotope values were obtained for seven anchialine species, all crustaceans, for the dry and rainy seasons. Among the species analyzed are primary consumers (atyid shrimps of the genus *Typhlatya*, the mysid *Antromysis cenotensis*), predators (the remipede *Xibalbanus tulumensis*, the amphipod *Tuluweckelia cernua*, the mysid *Stygiomysis cokei*) and a scavenger (the isopod *Creaseriella anops*). Six of the seven species had $\delta^{13}$C values that ranged between -37.8 ±3.1132 for *Typhlatya mitchelli* in the rainy season, to -24.0±1.264 for *Creaseriella anops* also in the rainy season (Table 3). These values are consistent with the feeding on organic matter from the outside that enters the cave. The second atyid species *T. pearsei* had very negative $\delta^{13}$C values (-53.2±1.9629 to -49.4±0.6) which can be explained by the consumption of organic matter derived from bacterial activity, specially methanotrophic bacteria (see Table 4). The $\delta^{15}$N values for six of the seven species ranged from 8.1±1.4882 obtained for *Typhlatya mitchelli* in the rainy season, to 14.7±0.3345 for *Xibalbanus tulumensis* in the dry season (Table 3). Clearly separated from this group is *Typhlatya pearsei* with extremely low values of -0.3 ±0.2376 and 1.4±0.4825, for the rainy and dry seasons, respectively (Table 3, Fig 4).

The $\delta^{13}$C values differed among some species, between the rainy and dry seasons and between seasons. Two primary consumers (*Typhlatya mitchelli*, *T. pearsei*), one predator (*Xibalbanus tulumensis*) and one scavenger (*Creaseriella anops*) showed marked differences between seasons with values differing from -4 to -5 in $\delta^{13}$C; whereas the species that had a similar behavior in both seasons (< -3 in $\delta^{13}$C) were one primary consumer (*Antromysis cenotensis*) and two predators (*Stygiomysis cokei*, *Tuluweckelia cernua*).

**Table 2. Total carbon in sediments (organic and inorganic fraction) through a transect inside the aquifer, this transect begins at the cenote pool and is directed to the cave region.** Sampling was carried out every 15 m until completing a distance of 75 m.

|  | Total carbon in sediments (%) | Organic carbon in sediments (%) | Inorganic carbon in sediments (%) |
|---|---|---|---|
| Tropical forest floor | 27.01 | 26.7 | 0.24 |
| Cenote pool sediment | 20.94 | 19.68 | 1.26 |
| 15 m | 11.71 | 4.32 | 7.39 |
| 30 m | 11.6 | 0.33 | 11.3 |
| 45 m | 11.39 | 2.86 | 8.53 |
| 60 m | 11.05 | 1.04 | 10.01 |
| 75 m | 10.46 | 0.44 | 10.02 |

**Table 3. Mean values (± SD) of δ$^{13}$C and δ$^{15}$N of seven anchialine species collected in Cenote Vaca Ha, first value corresponds to the rainy season and the second one to the dry season, n is the number of samples analyzed.**

| Species | δ$^{13}$C | δ$^{15}$N | n |
|---|---|---|---|
| *Antromysis cenotensis* | -33.5±0.0565 | 13.0±1.1234 | 6 |
| | -31.37±0.3240 | 11.2±0.0012 | 6 |
| *Stygiomysis cokei* | -28.9±0.1565 | 13.5±0.0565 | 2 |
| | -31.1±0.0587 | 14.6±0.1873 | 4 |
| *Typhlatya mitchelli* | -37.8±3.1132 | 8.1±1.4882 | 3 |
| | -33.60±0.4 | 11.50±0.053 | 10 |
| *Typhlatya pearsei* | -53.2±1.9629 | -0.3±0.2376 | 2 |
| | -49.4±0.6 | 1.4±0.4825 | 3 |
| *Creaseriella anops* | -24.0±1.264 | 10.5±0.7758 | 2 |
| | -32.8±1.2 | 8.2±0.4305 | 2 |
| *Tuluweckelia cernua* | -30.4±0.9627 | 12.7±0.2157 | 3 |
| | -30.6±0.2341 | 11.7±0.3468 | 4 |
| *Xibalbanus tulumensis* | -35.9±0.3223 | 12.8±0.0234 | 2 |
| | -30.9±0.1987 | 14.7±0.3345 | 2 |

**Table 4. Mean δ $^{13}$C values ± standard deviation for additional possible sources of contribution to the diet of consumers in Cenote Vaca Ha.** POM = particulate organic matter.

| | Dry Season | Rainy Season |
|---|---|---|
| Source | δ $^{13}$C | δ $^{13}$C |
| POM in the cave | -22.23±0.1915 | -22.419±0.5066 |
| Cave sediment | -13.7564±0.3456 | -27.6632±0.2916 |
| Nitrifying bacteria | -39.8333±3.4302 | -39.8333±3.4302 |
| [1] Methane oxidizing bacteria | -64.3633±5.4242 | -64.3633±5.4242 |
| [2] Methane oxidizing bacteria | -68.3666±5.4271 | -68.3666±5.4271 |
| AC (*Antromysis cenotensis*) | -31.3157±1.2211 | -32.8092±0.6088 |
| CA (*Creaseriella anops*) | -32.8±1.2211 | -24.0±1.2647 |
| TM (*Typhlatya mitchelli*) | -33.60±0.4 | -37.8±3.1132 |

[1] Reference value taken in Cenote Bang ~ 5 km away from Cenote Vaca Ha [30].

[2] Reference value from a compilation presented by [31].

The Bayesian mixing model results show the relative contribution of the most likely food sources to the diet of seven species considering the values of the isotopic signal and the isotopic fractionation (Fig 5, Table 4). The diet of the remipede *X. tulumensis* was largely based on the consumption of the mysid shrimp *A. cenotensis* and in a lesser proportion of the isopod *C. anops* and the atyid shrimp *T. mitchelli* in the rainy season; whereas in the dry season its diet was almost completely composed of *T. mitchelli* (Fig 5A); this is consistent with the differences observed in δ $^{13}$C between seasons. The mysid shrimp *Stygiomysis cokei* also shows a change in the diet between seasons, it consumed about 70% of *A. cenotensis*, 25% *T. mitchelli*, and 5% of *C. anops* in the rainy season, and about 95% the amphipod *T. cernua* and 5% *A. cenotensis* in the dry season (Fig 5B); however, the variation in δ $^{13}$C between seasons is only of -2.2 in δ $^{13}$C. The amphipod *Tuluweckelia cernua* shows a similar diet in both seasons consuming preferentially *T. mitchelli* and secondarily *C. anops* (Fig 5C), in this case the δ $^{13}$C values varied only -0.2. The diet of *T. mitchelli* is composed mainly of nitrifying bacteria in both seasons

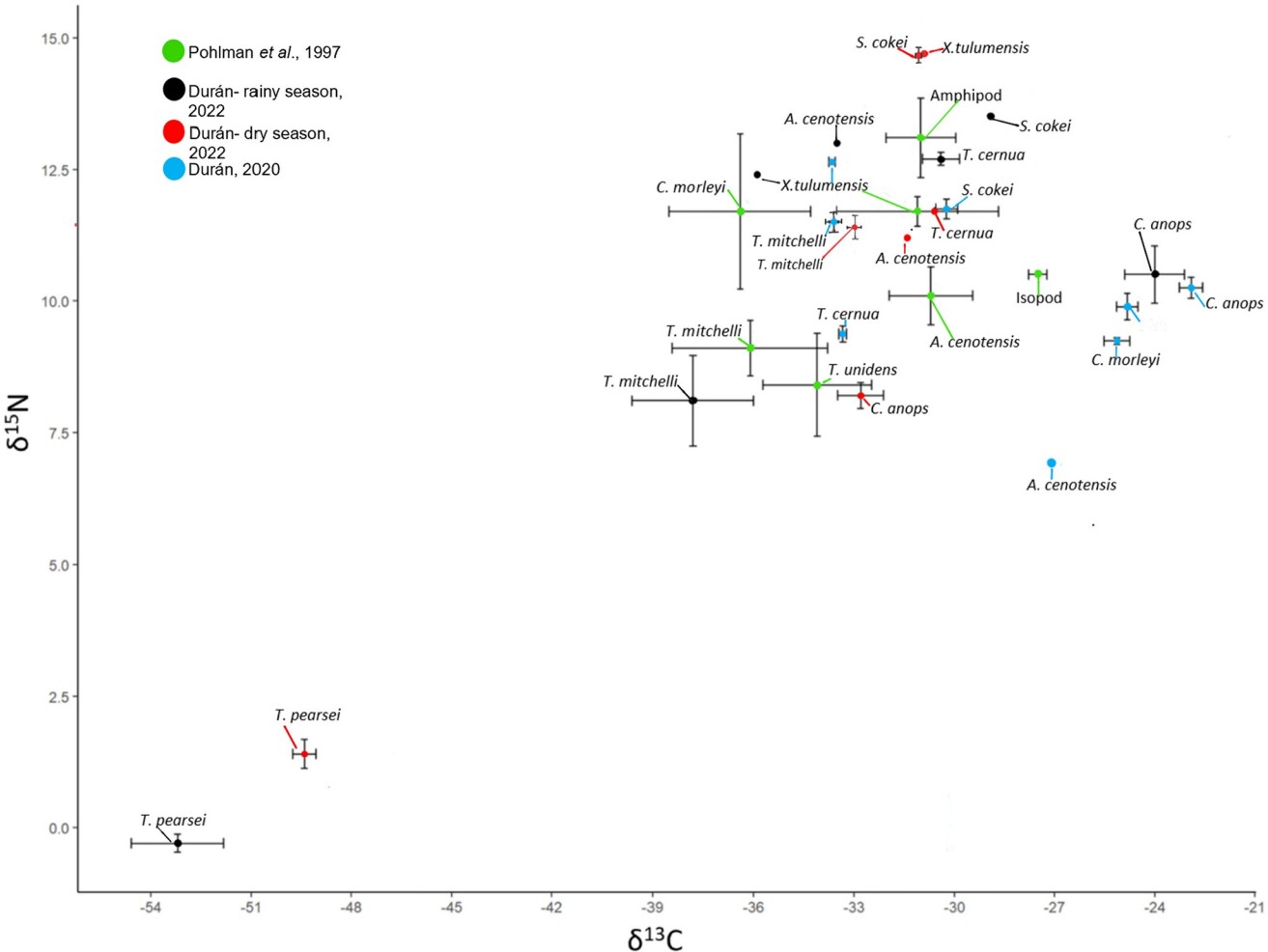

**Fig 4. Biplot of mean ± standard error of δ $^{13}$C and δ$^{15}$N values for four datasets: In green the one conducted from 1993 to 1995 [11], in blue the one from 2019 [23], and rainy (black) and dry (red) seasons from this study in 2022.** Shaded areas indicate the possible food source for the anchialine fauna.

with a reduced contribution of particulate organic matter and cave sediment (Fig 5D), but its δ $^{13}$C values varied -4.2. In the case of *T. pearsei* the feeding on methanotrophic bacteria could explain their low δ$^{13}$C values, with a variation of -3.8 between seasons (Fig 5E). The diet of the mysid *A. cenotensis* is consistent with the consumption of nitrifying bacteria and particulate organic matter in both seasons (Fig 5F). Finally, the cirolanid *Creaseriella anops*, shows a preference for cave sediment in the rainy season and a split preference for plant matter and the mysid *A. cenotensis* in the dry season (Fig 5G), which is reflected in a -8.8 difference in the δ$^{13}$C values between seasons.

To analyze the seasonal and interannual variation in the isotopic niche displayed by anchialine species we compared the results of this study with another three datasets using Bayesian ellipses (Fig 6). The four datasets show a considerable overlap (> 40% on average) for all species except for *T. pearsei* which due to very negative δ$^{13}$C values and very low δ$^{15}$N values lies outside the cluster formed by the rest of the anchialine species analyzed.

The trophic level analysis was not conducted as no reliable values of δ$^{15}$N could be obtained from the particulate organic matter in the water column or from the sediment in the cave, thus

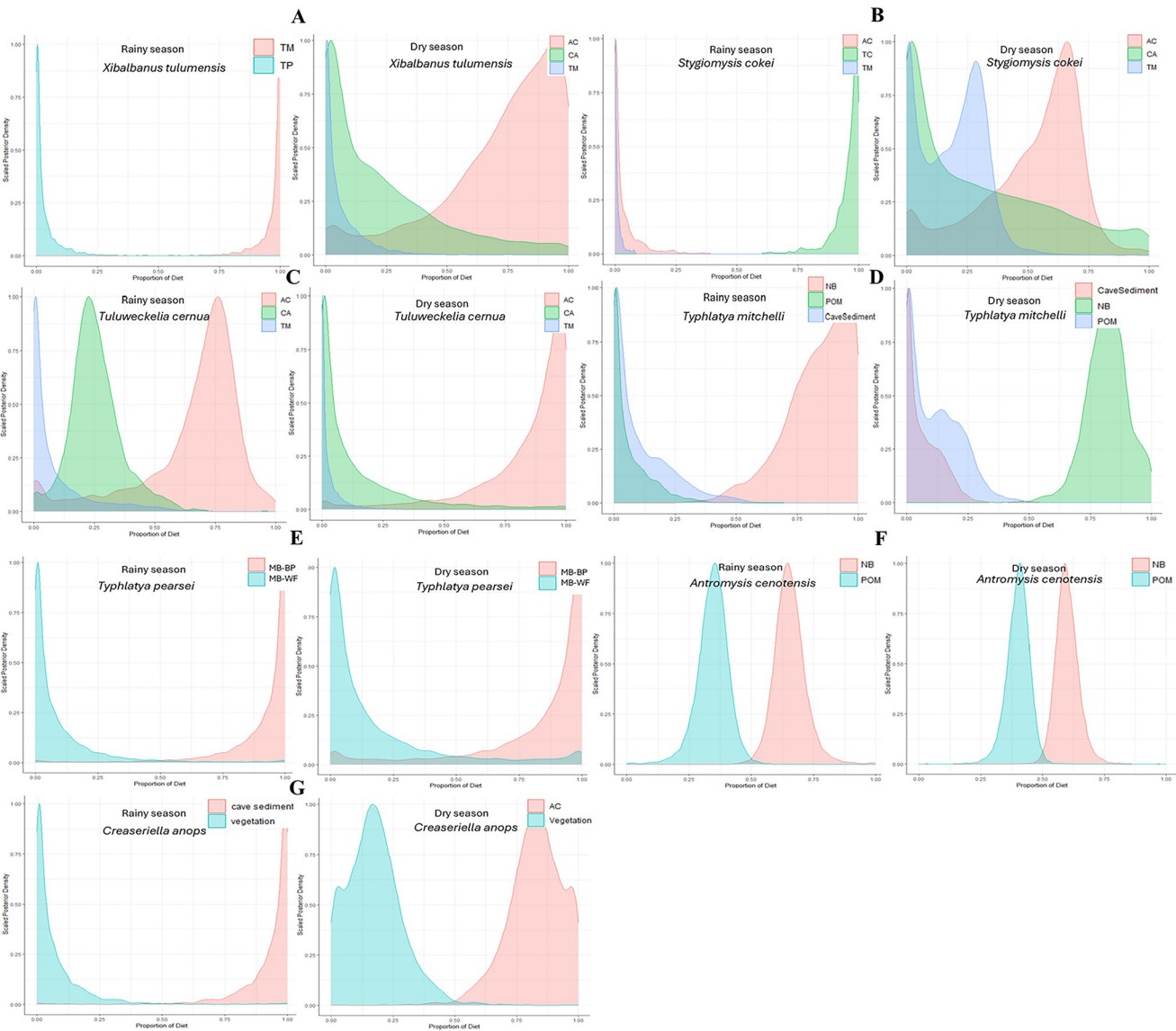

**Fig 5. Bayesian mixing model results to estimate the most probable diet sources.** (A) the remipede *Xibalbanus tulumensis*, (B) the mysid shrimp *Stygiomysis cokei*, (C) the amphipod *Tuluweckelia cernua*, (D) the atyid shrimp *Typhlatya mitchelli*, (E) the atyid shrimp *Typhlatya pearsei*, (F) the mysid shrimp *Antromysis cenotensis*, (G) the isopod *Creaseriella anops*. Abbreviations for the probable food sources are: AC, *Antromysis cenotensis*; CA, *Creaseriella anops*; TM, *Typhlatya mitchelli*; POM, particulate organic matter; NB, nitrifying bacteria; MB-BP, methanotrophic bacteria, data from [30]; MB-WF, methanotrophic bacteria, data from [31].

no primary production values were available to conduct the calculations. The N concentrations were below detection by the mass spectrometer.

## Discussion

### Environmental conditions

The water column conditions inside the cave were similar between the rainy and dry seasons showing little variation. The stratification remained without variation, with a distinct layer inside the dome of the cave, similar to what has been called a shallow halocline [22], which in

our case shows changes in temperature, dissolved oxygen concentration and pH, but not in salinity; a homogeneous freshwater layer, from 5 to 21 m depth; and a well-defined halocline at 21–22 m depth. Dissolved carbon in the water column and total carbon in sediments showed a clear inverse pattern in the freshwater layer where organic carbon decreases advancing inside the cave and inorganic carbon increases, while small changes can be seen in the marine water layer due to the different dynamics that operate there. In lowland areas close to the shoreline the YP the aquifer is influenced by high permeability of the substrate and thus high infiltration rates of meteoric water and by the tidally influenced marine water intrusion below the freshwater lens. Conditions are generally stable, but can reflect meteorological events such as strong storms or hurricanes with short-period peaks in water level and flow velocity [22, 32].

## Food web structure

Six of the seven anchialine species assessed from Cenote Vaca Ha form a well-defined cloud with $\delta^{13}$C values that reflect the consumption of organic matter transported to the cenote from the rainforest floor. Both, the POM in the cave and the cave sediment have isotopic values consistent with the input of organic matter from the exterior. Several authors have obtained similar isotopic values form anchialine fauna in the YP [16, 33]. Interestingly, the comparison of the four datasets considered, from 1993 through 1995 [11], 2019 [23] and this study including the 2022 rainy and 2023 dry seasons, that is a span of 30 years, shows a substantial overlap of the anchialine species isotopic niches. Temporal variation in isotopic niches has been widely documented at various scales, from weekly intervals to multi-year periods [34]. Being that stable isotopes reflect the organism's diet in relatively short periods of time [35] it is remarkable that the anchialine animal community in two nearby caves, Cenote Mayan Blue [11] and Cenote Vaca Ha ([23] and this study) show such a reduced amount of variation through time. Some species, like the shrimp *T. mitchelli*, a grazer and primary consumer, remained within a narrow range of variation in both isotopic proportions (-33.0 to -37.8 $\delta^{13}$C, 8.1–11.5 $\delta^{15}$N) through time, whereas others, such as the predatory palaemonid shrimp *Creaseria morleyi* showed a wider range of variation (-25.13 to -36.4 $\delta^{13}$C, 9.24–11.7 $\delta^{15}$N).

The case of the atyid shrimp *Typhlatya pearsei*, which has very negative $\delta^{13}$C values (-49.4 to -53.2‰), is interesting as these results place the species separate from the main isotopic space where the rest of the species overlap (Fig 6). Clearly, *T. pearsei* has a different diet, possibly composed, at least partially, of methanotrophic bacteria (Fig 5E and 5K), than those of the rest of the anchialine species which could be related to a spatial segregation or a different feeding behavior. In contrast to *T. mitchelli*, *T. pearsei* occurs in more internal sections of the cave where nutrient input from the exterior is limited and consequently could be consuming a biofilm composed by methanotrophic bacteria which have extremely low $\delta^{13}$C values [30, 31]. This scenario is consistent with the chemoautotrophic route proposed described from the nearby Ox Bel Ha cave system [21]. Although *Typhlatya* shrimp, due to their filtering/grazing feeding mode, were the key link assumed to be responsible for the introduction of chemoautotrophically derived organic matter into the food web, the question remained as to which one of the three possible species of *Typhlatya* was involved in this process. Existing low $\delta^{13}$C values for *T. pearsei* (-35.0 ± 1 to -41 ± 1‰) and *T. dzilamensis* (-31.0 ± 1 to -44 ± 0.6‰) recorded from different caves and times [16], lack a context to allow for a comparison. The values obtained for *T. dzilamensis* [16], which generally occurs below the halocline, were not explained fully since the methane concentrations as well as methanotrophic bacteria are present preferentially in the freshwater lens [21].

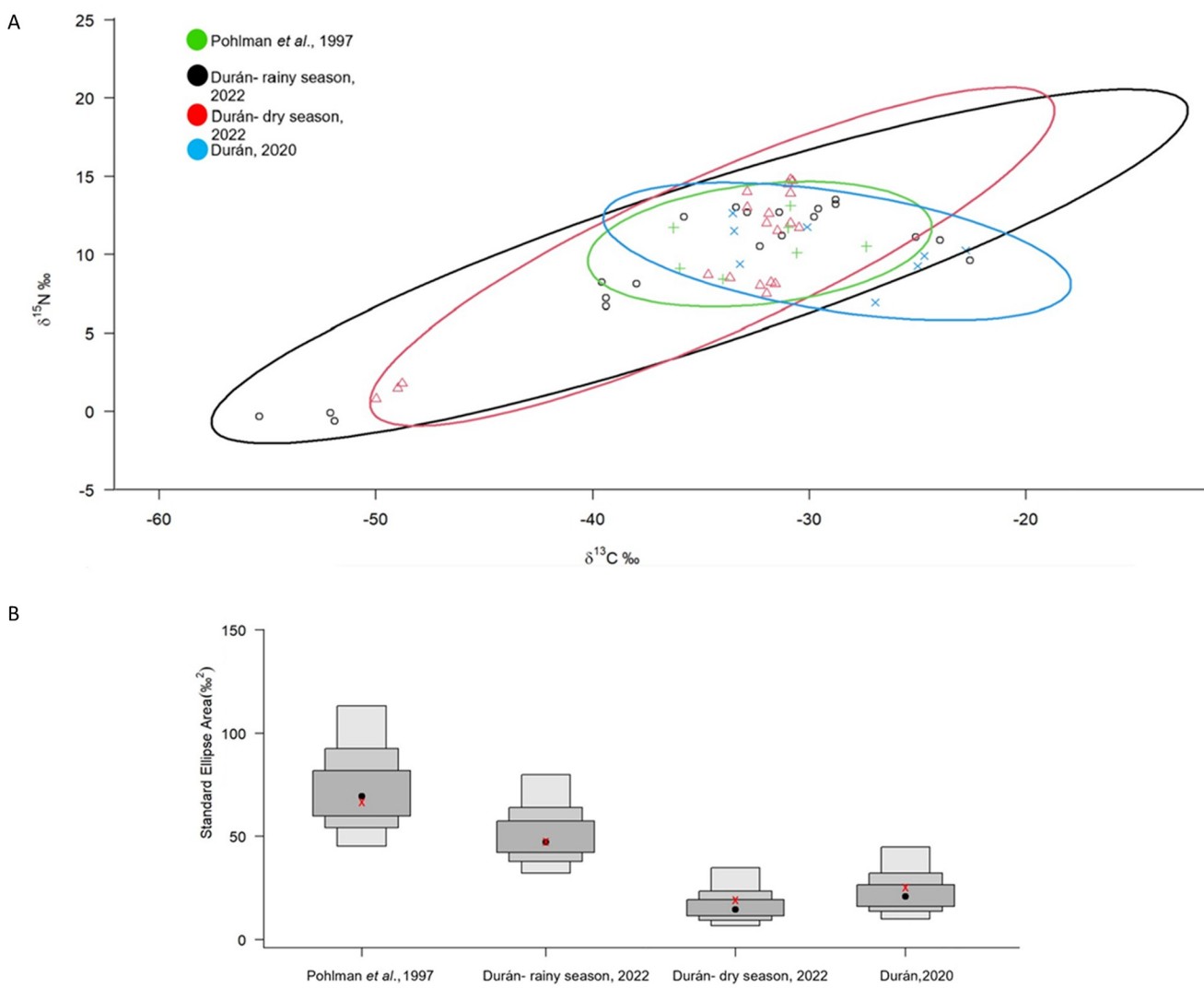

**Fig 6. Comparison of the isotopic niche of anchialine species.** (A) Bayesian standard ellipses comparing the isotopic niche of anchialine species based on four data sets [11, 23] and from the rainy and dry seasons. (B) density plots of Bayesian estimates of Standard Ellipse Areas (SEA) for the same datasets. Black dots indicate the modes of SEA, and boxed areas—Bayesian 50, 75 and 95% credible intervals; the red "x" indicates the mean.

Although any of the three possible species of *Typhlatya* present in the area could be feeding on methanotrophic bacteria, since they all have similar feeding apparatuses [18], here we propose, based on our findings and previously reported values of δ¹³C for *T. mitchelli*, that it is *T. pearsei* the key species that links the microbial processing of methane with the rest of the anchialine food web. *Typhlatya pearsei* is relatively rare, despite its wide distribution throughout the YP it always occurs in reduced numbers. In contrast, *T. mitchelli* is very common occurring from the surface to ~18 m depths in most caves that have been explored [36]. The rareness of *T. pearsei* could be very significant if it in fact represents one of the necessary components for the chemosynthesis-based food web to operate. The other necessary conditions are: rainfall, a well preserved rainforest on the surface to produce the organic matter that will decompose in the carbonate rock matrix, a cave conformation that allows for the formation of an anoxic freshwater layer next to the cave´s ceiling with methanogenic bacteria, the presence

of methanotrophic bacteria in a contiguous oxygenic layer that will use methane to synthesize organic matter, and the presence of *Typhlatya* shrimp to feed on this microbial mat introducing new organic matter into the food web. As we pointed out, the shrimp responsible for the last crucial step is *Typhlatya pearsei*, at least in the Caribbean cave area of Mexico [37]. We consider only *T. pearsei* for this role based on the very negative $\delta^{13}C$ values it shows and on the fact that this species is morphologically suited to graze on the bacterial mat that forms inside the cave. When more similar data become available, we will be able to determine if there are other species which could be occupying this role.

The tropic level of species was not determined since no reliable $\delta^{15}N$ base values could be obtained. Typically, $\delta^{15}N$ of the particulate organic matter in the water column or from the sediment is used as the base value, where bacteria, protists, algae, decaying plant material, and other small animals constitute a proxy to what could be considered the product of primary production. However, in this case N levels were too low to be detected. Nitrogen sources can be hard to identify as they depend on complex biogeochemical processes, microbial activity, amount of rainfall, and other direct sources [38, 39] and individuals of the same population may even show significant variation [40]. In this case the trophic roles of the studied species can be inferred from their morphology with support from observations in the field. *Typhlatya* shrimp have modified pereopods to filter feed or graze on the biofilm [18]; the mysid shrimp *Antromysis cenotensis* is usually seen at shallow depths or near the surface feeding on small POM particles; the remipede *Xibalbanus tulumensis* is a carnivore, it was recognized as the first venomous crustacean in the world [41]; the palaemonid shrimp *Creaseria morleyi* is also a predator although it can be an opportunistic omnivore [4]; the cirolanid isopod *Creaseriella anops* is a scavenger [19]; and both the amphipod *Tuluweckelia cernua* and the mysid shrimp *Stygiomysis cokei* due to their size and type of buccal appendages are clearly carnivores. Thus, we can infer the trophic role of these species based on their morpho-functional characteristics and $\delta^{13}C$ signatures.

## Seasonal variation in isotopic niche

SIA has been used to determine changes through time in the diet or trophic position since organisms incorporate, in relatively short periods of time, the isotopic signature of their food items [42, 43]. In various instances differences in isotopic values are the result of migrations or marked changes in the physical conditions of the habitat. We observed significant differences in the $\delta^{13}C$ values between the rainy and dry seasons in four out of the seven species analyzed. Water movement inside caves can be significant as freshwater slides towards the sea on top of a saline water intrusion that penetrates inland. Seasonal changes due to rainfall or strong tides create a very dynamic system specially in open conduits, as caves are, and more so in areas near the coastline [44]. Changes in salinity, temperature, conductivity, water level and other variables that describe the water column have been recorded in several sites along the Caribbean cave area [22, 32, 45]. However, despite the variations in the physical environment anchialine species seem, through time, to be feeding on the same sources producing similar isotopic signatures.

The small variations observed in this study of the isotopic values suggest the stability of the anchialine ecosystem at least in terms of nutrient input. Seasonal changes in the hydrological conditions of a cave due to rainfall in the same area as that of our study site were reported [22] showing that nutrient concentrations would decrease after strong rainfall due to a higher flow that would disperse them. So even if fluctuations in nutrient concentrations are weather-driven, their origin and composition remain relatively stable.

The obtained results show an anchialine ecosystem that depends on the nutrients generated by the rainforest that develops on the surface influence area of the cave. Nutrients can enter via

POM that collects in the cenote pool or through bacterial activity that transforms a variety of compounds into nutrients. In either case, the decaying organic matter from the rainforest is the primary food source for the anchialine species. Our results also show that the isotopic niches have only small variations with a substantial overlap of the community's isotopic space over time. In a sense, the predictability of the trophic behavior of the anchialine community renders it a very vulnerable one that could collapse as the surface environment is modified. We also foresee that the disappearance of key species, such as the shrimps of the genus *Typhlatya*, could draw with them complete microbial loops that are unique to this ecosystem, including the described chemosynthetic processing of methane. Refinements on the description of the anchialine trophic web are forthcoming with the use of modern sampling and analysis techniques.

## Supporting information

**S1 Table. Vertical profile Vaca Ha (dry-rainy season).** Water column physico-chemical variables taken with a multiprobe water-quality sonde for the two seasons.
(XLSX)

**S2 Table. Isotopic values Vaca Ha (dry-rainy season).** Values for $\delta^{13}C$ and $\delta^{15}N$ of all collected species for the two seasons.
(XLSX)

## Acknowledgments

Field work was possible thanks to the help and assistance of O. Cortés, A. Mora, A. Ceballos, B. Espinosa, M. Vázquez, J.L. Villalobos and P. Kovac. We thank D. Planas, A. Adamowics and J.F. Hélie for their assistance during the stable isotope analysis at the Centre de recherche Geotop, Université du Québec à Montréal. We thank J.C. Durán for his help with the elemental analyses of carbon and nitrogen in sediment and water at the Institute of Applied Science and Technology, UNAM.

## Author Contributions

**Conceptualization:** Brenda Durán, Fernando Álvarez.

**Data curation:** Brenda Durán, Fernando Álvarez.

**Formal analysis:** Brenda Durán, Fernando Álvarez.

**Funding acquisition:** Fernando Álvarez.

**Investigation:** Brenda Durán, Fernando Álvarez.

**Methodology:** Brenda Durán, Fernando Álvarez.

**Project administration:** Fernando Álvarez.

**Writing – original draft:** Brenda Durán, Fernando Álvarez.

**Writing – review & editing:** Brenda Durán, Fernando Álvarez.

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
