## [Decision Letter · Decision Letter 0]

13 Aug 2024

PONE-D-24-24608Trophic ecology in an anchialine cave: a stable isotope studyPLOS ONE

Dear Dr. Alvarez,

Thank you for submitting your manuscript to PLOS ONE. After careful consideration, we feel that it has merit but does not fully meet PLOS ONE’s publication criteria as it currently stands. Therefore, we invite you to submit a revised version of the manuscript that addresses the points raised during the review process.

We look forward to receiving your revised manuscript.

Kind regards,

Giorgio Mancinelli, Ph.D.

Academic Editor

PLOS ONE

Journal Requirements:

2. Thank you for stating the following financial disclosure: "BD CONAHCYT graduate scholarship 

FA grant PAPIIT IN206523 (DGAPA-UNAM) 

FA grant CONAHCYT Ciencia Básica A1-S-32846".

3. Thank you for stating the following in the Acknowledgments Section of your manuscript: "The first author thanks the Posgrado en Ciencias Biológicas, UNAM and CONAHCYT for the graduate scholarship awarded. We gratefully acknowledge the funding received through grants PAPIIT IN206523 (DGAPA-UNAM) and CONAHCYT Ciencia Básica A1-S-32846."

Please remove any funding-related text from the manuscript and let us know how you would like to update your Funding Statement. Currently, your Funding Statement reads as follows: BD CONAHCYT graduate scholarship 

FA grant PAPIIT IN206523 (DGAPA-UNAM) 

FA grant CONAHCYT Ciencia Básica A1-S-32846".

4. We note that your Data Availability Statement is currently as follows: "All relevant data are within the manuscript and its Supporting Information files."

6. Please upload a new copy of Figure 5 as the detail is not clear. Please follow the link for more information: "" ext-link-type="uri" xlink:type="simple">https://blogs.plos.org/plos/2019/06/looking-good-tips-for-creating-your-plos-figures-graphics/""
"" ext-link-type="uri" xlink:type="simple">https://blogs.plos.org/plos/2019/06/looking-good-tips-for-creating-your-plos-figures-graphics/""

7. We note that Figure 1 in your submission contain [map/satellite] images which may be copyrighted. All PLOS content is published under the Creative Commons Attribution License (CC BY 4.0), which means that the manuscript, images, and Supporting Information files will be freely available online, and any third party is permitted to access, download, copy, distribute, and use these materials in any way, even commercially, with proper attribution. For these reasons, we cannot publish previously copyrighted maps or satellite images created using proprietary data, such as Google software (Google Maps, Street View, and Earth). For more information, see our copyright guidelines: http://journals.plos.org/plosone/s/licenses-and-copyright.

Additional Editor Comments:

While two of the three reviewers found the quality of the manuscript generally good and required only a minor revision in order to be accepted for publication, a third reviewer highlighted a number of crucial flaws suggesting for a rejection. Personally, I find most of the criticisms made by rev #3 appropriate, even though my opinion is that they do not motivate a rejection. Accordingly, I recommend the authors a major revision of the ms integrating all the minor points raised by rev#1 and #2, and thoroughly considering and accounting for most, if not all the criticisms made by rev#3.

Reviewers' comments:

Reviewer's Responses to Questions

**Comments to the Author**

1. Is the manuscript technically sound, and do the data support the conclusions?

Reviewer #1: No

Reviewer #2: Yes

Reviewer #3: Yes

2. Has the statistical analysis been performed appropriately and rigorously? 

Reviewer #1: No

Reviewer #2: Yes

Reviewer #3: Yes

3. Have the authors made all data underlying the findings in their manuscript fully available?

Reviewer #1: No

Reviewer #2: Yes

Reviewer #3: Yes

4. Is the manuscript presented in an intelligible fashion and written in standard English?

Reviewer #1: Yes

Reviewer #2: Yes

Reviewer #3: Yes

5. Review Comments to the Author

Reviewer #1: The manuscript explores the trophic relationships among species occurring in an anchialine cave. The topic is surely of interest as very little is known. However, the manuscript has several lacks that hamper its further consideration. First, I would suggest the authors to rewrite it and pay attention to the linearity of their writing and to do not forget important information and concepts. Secondly, although the data is limited (a single cave, two seasons) it is anyway of high importance. Unfortunately, data are not analyzed in a proper way and also authors aim to answer unappropriated hypotheses. For example, collecting data from two different seasons is not enough to obtain information on year variability in carbon availability. This lack cannot be fixed using data from a different cave, which may be characterized by different dynamics. I feel that the data collected by the authors has potential, but the manuscript should be surely rewritten narrowing the focus on what can be clearly done.

Reviewer #2: Reviewer comments

The authors present an interesting study on the trophic network of cave fauna in the Yucatan Peninsula using stable isotopes, and the possible sources of energy going into the system. They evaluate seasonal changes in delta-carbon 13 and compare their results with those of two previous studies carried out in the same area. Their findings confirm the importance of the tropical forest surrounding the entrance to the caves, together with that of Typhlatya pearsei as key participants introducing organic matter into the short and oligotrophic webs that characterise flooded caves. The study highlights the use of stable isotopes in subterranean ecological studies and its conclusions contribute to the awareness of these fragile ecosystems.

The aims and hypotheses stated by the authors are clear; the methods used robust and the discussion is interesting and fully and adequately referenced. There are, however, some sections where the English writing could be improved (see some examples below), and I have only a few minor comments that might increase the clarity of procedures and enhance the visual communication of the results:

1) Some basic information on samples and sampling procedure is lacking.

• Line 173, provides the description of the way in which sediment and vegetation samples were processed to determine SIA. However, there is no previous mention of which and how vegetation samples were taken.

• It is not clear how many individuals of each species conformed a single sample: whether it was a single individual in every case, or individuals were pooled to conform a sample.

• It is not clear how many water samples were taken.

2) Table 1. It is unclear why there are no dispersion measures if there were several samples taken. Or was it only one water sample in each section of the aquifer?

3) Table 2. Use “tropical forest” instead of “jungle”. According to the Mat Met, there were more than one sample of sediment taken, but there are no dispersion values.

4) Table 3: Is n the number of individuals of each species in each season? or is it the number of samples with pooling of several individuals, particularly the smallest ones? This information needs to be provided, not only for the sake of repeatability, but because results should be interpreted in the light of that information.

5) Figure 4. A suggestion would be to change the colouring of the different samples in Figure 4: use letters to identify names of authors or studies (only have four different ones: Pohlman, Duran, rainy and dry seasons (present study); use colours for different species or groups of species according to feeding habits. This will make it easier to visualize samples pertaining to similar trophic levels and food sources.

6) Table 4 shows that the two-way ANOVA applied to the delta carbon data was strongly unbalanced; most surely it has high heterogeneity of variance. When there is a violation of the ANOVA assumptions, the results of the statistical tests are unreliable. Rather than going into trying to conform to these assumptions, I would recommend asking whether all of these tests are needed (7 Tukey tests + at least 1 F-test for the interaction term) only to conclude that delta-carbon 13 values differ between seasons in some species, but not in others? Furthermore, there is no ecologically relevant pattern to such differences (e.g. filter feeders differ between seasons, but predators don’t). Wouldn’t it be equally informative to merely describe differences in terms of the statistical parameters of the various samples (means, medians, dispersions) and avoid insisting on a statistical test that produces unreliable results? Take for example the p value of 1 for the Tukey test on Tuluweckelia cernua…such a result means that the null hypothesis is 100% certain… that makes no sense!

7) Line 298: One would expect that the species in which delta-carbon values changed among seasons (ANOVA results), would also show differences in the contribution of the various food sources from the rainy to the dry season in the Bayesian mixing models.... Was that the case? If not, why? To briefly compare these results would help consistency and integration of all statistical methods used in the study.

8) Figure 5. Legends and axis information is too small and the graphs are extremely difficult to read. Also, I would suggest the graphs to be ordered in such a way to facilitate the comparisons that are described in the text: between seasons, within each species. So, maybe show graphs of the two seasons for each species side by side.

9) Because Bayesian mixing models (as Bayesian statistics in general) are uncommon, a brief description of what these models attempts to achieve would improve the clarity of this section and allow the readers to better interpret these results.

10) From the Results section onward, there is a bit of confusion between the results obtained in the present study and those from previous works. This is particularly true in Table 6. This is an important aspect that needs to be attended, since it involves the analysis and discussion of changes in delta carbon values through different scales, and the conclusions drawn from them. For instance, were the previous and present studies carried out in the same seasons? If not, would the marked differences between seasons invalidate the comparison amongst years?

11) Line 421: Is T. pearsei the one and only species in that role? Are there no other candidates? It would be interesting to present and discuss other possible candidates. Maybe there is one that was not considered or sampled by the authors in the present study.

Some examples of language use that can be improved:

• Line 18: It should say “flows”

• Line 80: Use “Despite” instead of “Although”

• Line 107: It should say “Secondly, what species are involved in this process and how can their distribution affect such biogeochemical process…

• Line 115: Maybe change the phrase to “…possible to detect these changes to elucidate how the anchialine ecosystem reacts to them.”

• Line 153: Use approximately instead of “about”, and separate this phrase from the next using a semicolon.

• Line 220: Temperature was “near” 25.9 ºC.

• Line 225: Because Profiles are not statistically analysed, I should be better to use a word different from “significant”, given its strong statistical connotation.

• Line 360-365: This sentence is too long and difficult to read and understand. Consider re-writing it.

Reviewer #3: The manuscript presents a very valuable study focused on the identification the routes through which energy flow and to define the trophic network and niches of seven stygobiotic crustaceans in an anchialine cave Cenote Vaca Ha near Tulum in Quintana Roo, Mexico. It is based on analysis of carbon and nitrogen stable isotopes (δ13C and δ15N). Since the anchialine ecosystem is one of the less explored aquatic ecosystems in the world, this study is thus very original. Especially cenotes in Yucatán Peninsula connected with sea water are very suitable for such investigations. In addition to stygobiotic crustaceans, sediment, water and vegetation samples were analysed. The recent data were compared with two previous studies, both based on the identical species. This important contribution presents results on trophic niches of the seven crustaceans and observed that the shrimp Typhlatya pearsei is the key species to link chemosynthetic microbial production of organic matter to the anchialine trophic web. The results also showed that the isotopic niches had only small variations with a substantial

overlap of the community’s isotopic space over time. The authors observed substantial differences in animal isotope data between rainy and dry seasons.

The manuscript is very well written and concise. The study goals are clearly defined, used methods of data collection and analysis are appropriate. Combination of recent, new data with the previous observation allowed evaluation of the community trophic niches across several years.

I found two shortcomings to deal with:

1) low quality of uploaded pictures and so it was difficult to follow their meaningful value, especially Fig. 3 and Fig. 5,

2) it is necessary to stress in Introduction that nutrient scarcity is characteristic of deep cave zones and served as natural force for evolution of troglo-/stygobiotic animals, using relevant source(s), such as Encyclopedia of Caves, Cave Ecology (2018) or other basic cave biology monographs.

I found also several corrections to deal with further, all listed below.

I consider this manuscript as suitable for publication in PLoS ONE after minor revision.

Corrections

Abstract, line 17-18

Correct „energy flow“ to „energy flows“

The same in Introduction, line 37

Introduction, lines 80-85

Reformulate this text, since hardly to read. Try to divide the long sentence to make text more understandable.

Introduction, lines 108-109

Reformulate the text: “Third, do these processes influence the distribution of species creating.....“

Introduction, line 120

Reformulate the text: “….from the cave identical with the present study, obtained two years before the recent study (Durán 2020).“

Materials and Methods, lines 131-132

Reformulate the text: “The interior of the cave contains wide passages, enormous columns...“

Materials and Methods, lines 201-203

The following sentence should be associated to a respective figure or a table: “The results are presented as credibility intervals (50, 75 and 95) that describe the range of possible contributions from a particular source.

The same for the following paragraph of text.

Discussion, lines 368-369

Reformulate text: “…the consumption of organic matter transported to cenote from the rainforest floor.“

Line 370 – change “entering” with “input”

Line 471 – It would be better to write as: “…is the primary food source for the anchialine species.“

Table 1, lines 244-245

Reformulate the text: „the samples had a volume of 100 ml“

Table 2, lines 249-250

Formulate text in past tense.

Table 2, lines 251-252

Abbreviations were not used in the table, thus it is incorrect to provide them (TC, OC, IC) – erase this part table caption.

Figures 6, line 343

Correct text to „the red “x” indicates the mean“

6. PLOS authors have the option to publish the peer review history of their article (what does this mean?). If published, this will include your full peer review and any attached files.

Reviewer #1: No

Reviewer #2: No

Reviewer #3: **Yes: **Ľubomír Kováč

---

## [Author Response · Author response to Decision Letter 0]

25 Oct 2024

Response to reviewers

1. We followed the style templates for “main body” and title, authors, affiliations”.

2. We have stated in the cover letter that “The funders had no role in study design, data collection and analysis, decision to publish, or preparation of the manuscript”.

3. We removed all funding-related text from the “Acknowledgements” section. We have now included the funding information in the cover letter.

4. We have uploaded two supporting information files to complete the analyzed in this study.

5. – (this was skipped in the editor’s letter).

6. An improved version of Figure 5 has been uploaded.

7. We have created a new map for Figure 1, which now contains only elements that can be used freely.

Reviewer 1 (marked copy).

Line 20. The term “unique” was changed for “endemic”.

Line 21. We added at the end of the sentence “analyzed to determine what the main nutrient sources are”, to complete the idea.

Line 23. We now specify that the seven species we studied are the same that were studied in the other two previous studies (line 33 in the new version).

Line 25. The reviewer asks what we mean by “anchialine niche”, now it reads “anchialine isotopic niche”, clarifying the idea.

Lines 25-27. Subsection “b” of this sentence now reads “. . . . confirm previous trophic classifications; . . . “. The idea is that based on morphology alone the studied species, which are endemic and rare, were assigned by different authors a hypothetical trophic role, our study contributes with the contribution models additional evidence of what the trophic role is for each one. 

Line 30. The reviewer asks what is original about our findings, I will refer to lines 23-30 of the marked copy. Subsection “a”, no previous study have compared the δ13C and δ15N values of anchialine fauna through time, which is a central point in our study; “b”, as mentioned above, anchialine species are rare and their trophic roles have been inferred from anatomical attributes and a few observations at the community level, now we analyze their trophic roles through contribution models based on stable isotope data; “c”, no previous study have proposed which of the anchialine species in the YP could be the main link in a microbial based food web, including methanotrophic bacteria, and thus here we present evidence that points to Typhlatya pearsei as the species that has this crucial role. 

Line 38. The word “visualize” has been changed for “identify”.

Line 40. The suggested reference “Mammola et al., 2021” was included in the sentence.

Line 48. The reviewer assumes that because our study site is in the Yucatan Peninsula the water in the caves have more organic resources (nutrients) compared to temperate regions. Water quality in the flooded caves of the YP has been many times considered as oligotrophic or even ultraoligotrophic in preserved areas (Álvarez et al., 2023. Anchialine Fauna of the Yucatan Peninsula: Diversity and Conservation Challenges. In: Jones, R.W., Ornelas-García, C.P., Pineda-López, R., Álvarez, F. (eds) Mexican Fauna in the Anthropocene. Springer, Cham. https://doi.org/10.1007/978-3-031-17277-9_13; Torres-Talamante et al., 2011, The key role of the chemolimnion in meromictic cenotes of the Yucatan Peninsula, Mexico. Hydrobiologia 677, 107–127 https://doi.org/10.1007/s10750-011-0746-9). Of course, intensive development in the region has contaminated to different extents specific sections of the aquifer as it occurs around the world, but large portions of the flooded caves network remain in good condition with very low nutrient concentrations (e.g., Metcalfe et al., 2011 – Environmental Pollution 159:991-997). In any case, the evolution of the anchialine fauna has occurred in low nutrient conditions in all anchialine caves around the world (Iliffe Alvarez, 2018 – Research in anchialine caves in O., Kováč, Ľ., Halse, S. (eds) Cave Ecology. Ecological Studies, vol 235. Springer, Cham. https://doi.org/10.1007/978-3-319-98852-8_18). 

The reviewer also states that this cave is “less divergent from the surface environment”. We do not know on what data is the reviewer founding this assertion. Our opinion and that of many authors is that the anchialine cave environment is unique and cannot be compared to the surface environment.

Line 55. The correction has been made, we changed “different” for “differently”.

Line 57. The sentence was deleted and incorporated in the last paragraph of the Introduction.

Line 64. The reviewer suggests that the definition of “cenote” here is not necessary because it was mentioned lines above. However, since we moved the sentence (line 57) where this was mentioned, now we feel that the definition is necessary.

Line 89. The reviewer comments: “you are jumping from one topic to the other and the reading is not linear”.

Paragraph from lines 36 to 44, topic – stable isotope analysis could be useful to study the anchialine trophic web

Paragraph from lines 45 to 59, topic – anchialine caves have a rich fauna and are characterized by low nutrient concentrations. 

Paragraph from lines 60 to 79, topic – previous studies of anchialine fauna in the Yucatan have described the possible trophic roles of several species.

Paragraph from lines 80 to 92, topic – SIA has identified that nutrients could also enter the anchialine environment through microbial activity.

Paragraph from lines 93 to 100, topic – from SIA data it is possible to link microbial activity to the trophic role of certain species.

Paragraph from lines 101 to 110, topic – recent studies that identify the existence of chemosynthesis in the anchialine environment support these results.

Paragraph from lines 111 to 120, topic – summary of relevant questions that we are addressing in this study.

This is the sequence of ideas that are dealt with in the introduction. We go from general to particular ideas; however, to clarify this sequence we added the following sentence at the end of the first paragraph: “In order to follow a logical sequence, we need to know what species occur in the anchialine habitat and what their likely trophic roles are, how nutrients are entering the caves and what the possible sources are, and finally under what degree of variation – spatial and temporal – is this structure operating? 

Line 111. We agree with the comment. However, we deem improbable that the anchialine cave environment could suffer the invasion of “allochthonous (epigean)” species due to an increased nutrient input. Rather, the most probable effect would be the loss of native species.

Line 117. We compare the results of this study with a previous one conducted in the same cave two years before and with another one conducted 18 years ago with the same species in a nearby cave. We are aware of the shortcomings of this comparison, but this the only other comparable study available. 

Line 141. We are aware of the limitations of the comparison. However, using a study conducted in the same way, with the same species, in a similar cave, only a few kilometers away, represents a good opportunity to gain knowledge about the scale of temporal variation in the trophic structure of this community.

Line 145. More detail on the sampling of organisms has been added (lines 188-189 in the new version)

Line 148. Up to 50 m around the cenote, this has been included in the text.

Line 148. There are fish in the cenote pool, mainly of the genus Astyanax, that eat all insects and other small animals that fall into the pool. Whereas leaves and other plant material sink and collect in the bottom undergoing a slow decaying process. During this time is when it can be transported into the cave.

Line 190. See comment number 6 under Reviewer 2.

Line 213. As is shown later in the section and in Figure 3, there is one complete water quality profile for each one of the samplings.

Line 256. The studied species are now mentioned in the last paragraph of the Introduction.

Line 260. We are just referring to Table 3, mentioning the species that have the maximum and minimum values for each isotope.

Line 284. See comment 6 of Reviewer 2 below.

Line 299. We are referring to Figure 5A, this is a result of our study.

Reviewer 2.

1. Line 173. The reviewer asks for more details on the collecting methods and sample sizes.

Vegetation samples consisted of leaves lying on the rainforest floor coming from the more common trees around the cenote. The most common species around the cenote are listed in the manuscript. 

More details of how the water samples were obtained were added. Three filtered subsamples were obtained from the pool, cave freshwater and cave salt-water.

Regarding the samples for SIA, we explain that in the case of small species (mysid shrimps and amphipods) several organisms were used to obtain the required sample for the analysis. With larger species (isopods, atyid and palaemonid shrimps, remipedes) one organism was used to obtain one sample.

2. Table 1. There are no dispersion measures because there was one water sample for each section of the system.

3. Table 2. There were 7 sediment samples taken and analyzed to cover the transect from the rainforest floor to 75 m inside the cave. The idea was to determine how the organic and inorganic carbon fractions varied from the outside to inside the cave.

4. Table 3. We now specify that “n” is the number of samples analyzed, not the number of organisms. We explained in Mat Meth that we used several organisms for one sample in case they were very small (mysid shrimps and amphipods) and that we used one organism per sample with large specimens. 

5. Figure 4. We have modified the figure to improve its clarity.

6. Table 4. The reviewer questions the usefulness of the ANOVA used to test for differences in isotopic values between species and between seasons. Since the number of values is limited due to the difficult conditions to sample this environment and the terms of the ANOVA cannot be improved, we follow his/her advice and have eliminated this analysis from the manuscript. Instead, we added more detail to the direct comparison of δ13C values.

7. Line 298. Yes, this is a very pertinent comment by the reviewer. We have now improved the paragraph where the results of the contribution models are presented. We added the information that allows the reader to know in which cases a change in the diet composition occurs with a change in the δ 13C values of the species. As is now presented, in some cases the δ 13C values change with the diet and in others it doesn’t.

8. Figure 5. The figure has now been improved as suggested.

9. Bayesian mixing models are described in the “Data analysis” section of Materials and Methods. We expanded this explanation and added a reference.

10. The reviewer comments on the data presented in Table 6 (Table 4 in the new version), which might be confusing. The isotopic values from other sources that are used for comparisons need a more detailed explanation. We have added notes to Table 4 to improve the clarity.

11. This is an interesting question by the reviewer. The reviewer asks if Typhlatya pearsei is the only species that has the role of consuming methanotrophic bacteria. Our data shows that in this cave it is the only species that can have such a role. We have added a couple of sentences in the Discussion to emphasize this point (lines 492-496 of the new version).

Line 18. Correction done.

Line 80. Correction done.

Line 107. Correction done (line 126 in the new version).

Line 115. Correction done (line 134 in the new version).

Line 153. Changes done (line 185 in the new version).

Line 220. Change done (line 269 in the new version).

Line 225. Change done (line 275 in the new version).

Lines 360-365. Sentence too long. The sentence has been divided to improve clarity (lines 435-438 in the new version).

Reviewer 3.

1. Figures 3 and 5 have been modified and improved. The rest of the figures have been checked

2. We have added a paragraph in the Introduction emphasizing the nutrient-poor condition of anchialine caves in the Yucatan.

Lines 17-18. Corrections done.

Lines 80-85. The sentence has been divided as it was too long and hard to read (lines 105-111 in the new version).

Lines 108-109. The sentence was modified as suggested (lines 135-136 in the new version).

Line 120. The sentence was modified as suggested (lines 155-156 in the new version).

Lines 131-132. The sentence was modified as suggested (line 168 in the new version).

Lines 201-203. The reviewer suggests that the description of the graphs used to represent the results of the Bayesian mixing models (BMM) presented in the “Materials and methods” section should be associated to a figure. When the results of the BMM are presented they are associated to a figure (paragraph starting on line 356 of the new version). In any case, we added a call to Fig 5 in that paragraph, and a call to Fig 6 in the next paragraph.

Lines 368-369. The sentence was modified as suggested (lines 455-456 in the new version).

Line 370. Change done (line 457 in the new version).

Line 471. Correction done (line 568 in the new version).

Table 1, lines 244-245. Correction made (line 307 in the new version).

Table 2, lines 251-252. Correction made (line 315 in the new version).

Figure 6, line 343. Correction made (line 432 in the new version).

---

## [Decision Letter · Decision Letter 1]

2 Dec 2024

Trophic ecology in an anchialine cave: a stable isotope study

PONE-D-24-24608R1

Dear Dr. Alvarez,

We’re pleased to inform you that your manuscript has been judged scientifically suitable for publication and will be formally accepted for publication once it meets all outstanding technical requirements.

Kind regards,

Giorgio Mancinelli, Ph.D.

Academic Editor

PLOS ONE

Additional Editor Comments (optional):

After a second revision round, the current version of the manuscript successfully integrates all the points raised by the reviewers and is now acceptable for publication in PLOS ONE.

Reviewers' comments:

Reviewer's Responses to Questions

**Comments to the Author**

1. If the authors have adequately addressed your comments raised in a previous round of review and you feel that this manuscript is now acceptable for publication, you may indicate that here to bypass the “Comments to the Author” section, enter your conflict of interest statement in the “Confidential to Editor” section, and submit your "Accept" recommendation.

Reviewer #1: All comments have been addressed

Reviewer #3: All comments have been addressed

2. Is the manuscript technically sound, and do the data support the conclusions?

Reviewer #1: Yes

Reviewer #3: Yes

3. Has the statistical analysis been performed appropriately and rigorously? 

Reviewer #1: Yes

Reviewer #3: Yes

4. Have the authors made all data underlying the findings in their manuscript fully available?

Reviewer #1: Yes

Reviewer #3: Yes

5. Is the manuscript presented in an intelligible fashion and written in standard English?

Reviewer #1: Yes

Reviewer #3: Yes

6. Review Comments to the Author

Reviewer #1: The authors made substantial revisions on their manuscript, addressing Reviewers concerns. I feel that now their manuscript is ready to be accepted. Congratulations.

Reviewer #3: I have revised the new version of the manuscript. Since the authors addressed all of my previous comments/edits/concerns, I consider it to be suitable for publishing in the journal Plos One.

7. PLOS authors have the option to publish the peer review history of their article (what does this mean?). If published, this will include your full peer review and any attached files.

Reviewer #1: **Yes: **Enrico Lunghi

Reviewer #3: **Yes: **Ľubomír Kováč

---

## [Editor Report · Acceptance letter]

6 Dec 2024

PONE-D-24-24608R1 

PLOS ONE

Dear Dr. Alvarez, 

I'm pleased to inform you that your manuscript has been deemed suitable for publication in PLOS ONE. Congratulations! Your manuscript is now being handed over to our production team.

Kind regards, 

on behalf of

Dr. Giorgio Mancinelli 

Academic Editor

PLOS ONE